# An Insight of *Betula platyphylla SWEET* Gene Family through Genome-Wide Identification, Expression Profiling and Function Analysis of *BpSWEET1c* under Cold Stress

**DOI:** 10.3390/ijms241713626

**Published:** 2023-09-04

**Authors:** Hao Zhang, Yuting Ding, Kaiye Yang, Xinyu Wang, Wenshuo Gao, Qingjun Xie, Zhongyuan Liu, Caiqiu Gao

**Affiliations:** State Key Laboratory of Tree Genetics and Breeding, Northeast Forestry University, Harbin 150040, China; haozhang@nefu.edu.cn (H.Z.); dingyuting@nefu.edu.cn (Y.D.); yangkaiye@nefu.edu.cn (K.Y.); wang_xinyu@nefu.edu.cn (X.W.); gws000211@nefu.edu.cn (W.G.); xieqj@nefu.edu.cn (Q.X.); liuzhongyuan@nefu.edu.cn (Z.L.)

**Keywords:** *Betula platyphylla*, SWEET, expression profile, cold, function analysis

## Abstract

SWEET proteins play important roles in plant growth and development, sugar loading in phloem and resistance to abiotic stress through sugar transport. In this study, 13 *BpSWEET* genes were identified from birch genome. Collinearity analysis showed that there were one tandem repeating gene pair (*BpSWEET1b*/*BpSWEET1c*) and two duplicative gene pairs (*BpSWEET17a*/*BpSWEET17b*) in the *BpSWEET* gene family. The *BpSWEET* gene promoter regions contained several cis-acting elements related to stress resistance, for example: hormone-responsive and low-temperature-responsive cis-elements. Analysis of transcriptome data showed that *BpSWEET* genes were highly expressed in several sink organs, and the most *BpSWEET* genes were rapidly up-regulated under cold stress. *BpSWEET1c*, which was highly expressed in cold stress, was selected for further analysis. It was found that *BpSWEET1c* was located on the cell membrane. After 6 h of 4 °C stress, sucrose content in the leaves and roots of transient overexpressed *BpSWEET1c* was significantly higher than that of the control. MDA content in roots was significantly lower than that of the control. These results indicate that *BpSWEET1c* may play a positive role in the response to cold stress by promoting the metabolism and transport of sucrose. In conclusion, 13 *BpSWEET* genes were identified from the whole genome level. Most of the *SWEET* genes of birch were expressed in the sink organs and could respond to cold stress. Transient overexpression of *BpSWEET1c* changed the soluble sugar content and improved the cold tolerance of birch.

## 1. Introduction

Sugar plays a significant role in the plant developmental process and abiotic stress responses [1,2,3]. In these biological processes, sugar is transported from the source organs over a long distance through the phloem to the sink organs, achieving the distribution of sugar within the plant body. Sugar transport includes three processes: sugar loading in phloem [4], vascular bundle transport and sugar unloading in phloem [5]. During these processes, a large proportion of sugar transport relies on sugar transport proteins. In the apoplast loading pathway of sugar in phloem, sugar depending on sugars will eventually be exported transporters (SWEETs) to transfer from phloem parenchyma cells to the nearby sieve tube–companion cell complex [6], and then transported by sucrose transporters (SUTs) to the sieve tube–companion cell complex [7]. In the process of sugar unloading in phloem, the sugar in the apoplast is decomposed into monosaccharide and then transported to parenchymal cells through monosaccharide transporters (MSTs) [8].

In contrast with SUTs and MSTs, SWEETs can bidirectionally transport sugar along a concentration gradient [9], independent of proton gradient. In addition, SWEET proteins can also transport various soluble sugars such as glucose and fructose [10]. SWEET proteins are a new class of sugar transport which was discovered in 2010 [9]. The most significant feature of SWEET proteins is that it has MtN3/saliva or PQ-loop conserved transmembrane domain [10]. SWEET proteins consist with 7 TMs (7α-helical trans-membrane domains), the C-end and N-end are two relatively conservative 3-TMs, which are helically connected by signal TM with lower conservation in the middle [9]. 

With the frequent occurrence of extreme weather phenomena, cold stress is causing greater harm to plants. When the cold stress is coming, the plant membrane system will be damaged. Sugar involves the process of plant defense against cold stress by stabilizing cellular components and membranes [11]. More and more studies have shown that SWEET proteins are involved in plant cold tolerance through sugar transport and metabolism. For example, 30 *BoSWEETs* have been identified in cabbage (*Brassica oleracea*). Among them, *BoSWEET11b*, *11c*, *12b*, *16a* and *17* were down-regulated when facing chilling stress. The genes down-regulation may cause the higher concentration of glucose and fructose and increase the tolerance of chilling [12]. Additionally, the vacuolar fructose exporter *AtSWEET17* responds to cold stress by transporting fructose [13]. Furthermore, *AtSWEET16*, the closest homolog of *AtSWEET17*, also play an important role in cold resistance by increase the amount of glucose and sucrose to compensate the loss of fructose [14]. Overexpression of *HfSWEET17* in tobacco enhances cold tolerance [15]. Taken together, these results suggest that *SWEET* genes play important roles in resistance to cold stress.

Birch (*Betula platyphylla*) is widely distributed in Northeast China, North China and other regions. As a pioneer tree species, birch has strong tolerance to cold stress and has developed a mature cold tolerance mechanism. Additionally, birch has a higher photosynthetic capacity and growth rate than other tree species [16]. Furthermore, birch can adapt to poor nutrient soils and dramatically improve the ecological environment. In recent years, with the completion of whole genome sequencing of birch, many molecular mechanisms of cold tolerance of birch have been revealed [17]. However, the function of SWEET proteins in birch in the cold stress response have not been noted. It is necessary to explore the *SWEET* gene family and the relation between *BpSWEETs* and cold tolerance.

In this study, the 13 *SWEET* genes were identified from birch. These *BpSWEET* genes were divided into four categories by constructing a *BpSWEET* evolutionary tree, and gene structures, chromosome localization and the cis-acting elements of promoter regions were analyzed. Through RNA-seq data, we found that *BpSWEET* genes have different expression levels in different organs and tissues, and most *BpSWEET* genes respond to cold stress. In addition, we conducted subcellular localization of a tandem repeating gene *BpSWEET1c* and found that *BpSWEET1c* was located in the cell membrane, and transient overexpressed *BpSWEET1c* may improve cold tolerance by changing the soluble sugar content.

## 2. Results

### 2.1. Identification and Evolutionary Tree Analysis of BpSWEET Gene Family of Birch

A total of 14 *BpSWEET* genes were identified from the birch genome database, one gene has been excluded because the deletion of the gene structure. The 13 *BpSWEET* genes were named as *BpSWEET1a* to *BpSWEET17b*, referring to their Arabidopsis homologs. Multiple sequence alignment showed that all 13 *BpSWEET* genes had two PQ-loop super family domains (Appendix A). The protein length of *BpSWEET* genes was calculated to range from 235 aa (BpSWEET2) to 313 aa (BpSWEET17b). The predicated molecular weight ranges from 26.18 kDa (BpSWEET2) to 35.49 kDa (BpSWEET17b), and theoretical isoelectric points (pl) varied from 5.79 (BpSWEET15) to 9.54 (BpSWEET1b). In addition, all proteins have seven transmembrane domains. Subcellular localization predicted that two SWEET proteins (BpSWEET9, BpSWEET17a) were located in the vacuolar membrane and other SWEET proteins were located in the plasma membrane (Table 1).

To better understand the evolutionary relationship of SWEET proteins, we constructed a phylogenetic tree by aligning 13 BpSWEET proteins from birch and 17 AtSWEET proteins from Arabidopsis. According to the phylogenetic tree, 13 BpSWEET proteins can be grouped into four clades (Clades I, Clades II, Clades III and Clade IV, Figure 1). Clades I include 5 BpSWEETs; Clades IV includes 4 BpSWEETs. Clades II and Clades III have fewer proteins, which includes 2 BpSWEETs. This result implied that the proteins from Clades II and Clades III may be conservative in evolution.

### 2.2. Gene Structures and Conserved Domains Analysis of BpSWEET Genes

Gene structures and gene motifs play an important role in the analysis of the characters and evolution of the gene family. The gene structures and conserved motifs of 13 *BpSWEET* genes were further analyzed (Figure 2). The results showed that 11 *BpSWEET* genes have 6 exons, *BpSWEET7* has 5 exons and *BpSWEET17b* has 7 exons. In addition, most genes (10 *BpSWEETs*) have five introns. However, *BpSWEET17b* and *BpSWEET2* have six introns, and *BpSWEET4* has four introns.

The results of conserved motifs analyzed by MEME showed that a total of ten motifs were identified in the *BpSWEET* gene family, which are named motif 1–10, respectively (Figure 2). All *BpSWEET* genes contain motif 1, 2, 3, 4, 5 and 7. A total of 8 *BpSWEET* genes, including *BpSWEET1a*, *2*, *3*, *4*, *7*, *10*, *11* and *15*, contain motif 10. Moreover, *BpSWEET15* and *BpSWEET17b* contain motif 9. Interestingly, three homologous genes of the *AtSWEET1* gene (*BpSWEET1a*, *BpSWEET1b* and *BpSWEET1c*) were all contain motif 6 at the C-terminal. Similarly, motif 8 at C-terminal was found in 2 homologous genes, *BpSWEET17a* and *BpSWEET17b*.

### 2.3. Chromosome Mapping and Collinearity Analysis of BpSWEET Genes

The chromosome mapping of 13 *BpSWEET* genes was analyzed (Figure 3). The result showed that the genes were distributed on seven chromosomes. Chr02 and Chr12 have the largest number of *BpSWEET* genes. Chr02 includes three genes: *BpSWEET7*, *BpSWEET10* and *BpSWEET11*. Chr12 includes three homologous genes: *BpSWEET1a*, *BpSWEET1b* and *BpSWEET1c*. Furthermore, Chr05 and Chr08 have two *BpSWEET* genes. Chr01, Chr04 and Chr11 contain only one *BpSWEET* gene. Interesting, one group tandem duplication gene pair (*BpSWEET1b*/*1c*) is located in Chr12. This result demonstrates that tandem duplication is partly responsible for the expansion of *BpSWEETs*.

A large part of gene duplication in plants comes from chromosome inter-chromosomal duplications. To learn the *BpSWEETs* evolutionary origins, the intraspecific synteny analysis of SWEET proteins in birch was performed (Figure 4A). According to synteny analysis, two pairs of segmental duplication genes were found (*BpSWEET11*/*15*, *BpSWEET17a*/*17b*), which locate on Chr01, 02, 05 and 08. We have marked the gene pairs in the figure with the red line. By synteny analysis, we inferred that one gene in each two gene pairs originates from the segmental duplication of the other gene.

To further explore the evolutionary relationship of *SWEET* genes among different species, we analyzed the homologs between birch and six other angiosperms, which includes five dicotyledonous plants (*Arabidopsis thaliana*, *Populus trichocarpa*, *Vitis vinifera*, *Amborella trichopoda* and *Solanum lycopersicum*) and one monocotyledonous plant (*Oryza sativa*) (Figure 4B). The results showed that *P. trichocarpa* has the most homologous genes with birch—with 20 pairs of homologous genes, followed by *S. lycopersicum* (13 pairs), *V. vinifera* (12 pairs), *Arabidopsis* (8 pairs), *A. trichopoda* (5 pairs), *O. sativa* (4 pairs). The result indicated that dicotyledonous plants, especially *P. trichocarpa*, have a high evolutionary homology with *SWEET* genes in birch.

Ka/Ks is widely used to measure the speed of gene evolution, reflecting the selection pressure in evolution [18]. The Ka/Ks results of *BpSWEET* paralogous genes showed the Ka/Ks ratio of three gene pairs (*BpSWEET1b/1c*, *BpSWEET11/15*, *BpSWEET17a/17b*) less than 1, suggesting that the genes have conservative function and have undergone purification selection (Appendix A). Furthermore, *SWEET* orthologous genes were selected from birch and Ka/Ks analysis was performed with the orthologous genes from other two species (*P. trichocarpa*, Arabidopsis) (Appendix A). In data, we can infer that these genes have undergone purification selection. They are also relatively conservative in function between birch and Arabidopsis and *P. trichocarpa* [19,20,21].

### 2.4. Analysis of BpSWEETs Promoter Regions

To better understand the biological processes involved in *BpSWEETs*, the cis-acting elements of the promoter regions of 2000 bp were analyzed (Figure 5). A total of 15 cis-acting elements were identified and selected by Plantcare. The elements mainly included: (1) cis-acting elements related to stress, containing low-temperature, drought and so on. (2) Cis-acting elements of corresponding hormones, such as abscisic acid, Me-JA, salicylic acid, gibberellin, auxin. (3) Cis-acting elements involved in growth and development. (4) Cis-acting elements that participate in the synthesis of secondary metabolites. (5) Cis-acting elements that control circadian rhythm. Overall, the results implied that *SWEET* genes involved many biological processes, play important roles in resisting stress, respond to hormones, and other biological processes.

Different subfamilies of the same gene family often have different functions. The functional differentiation of different subfamilies was analyzed from the perspective of cis-elements (Figure 5). A large part of the Class I family contains Me-JA responsive and low-temperature responsive cis-elements. Among the five members of the Class I family, four contain low-temperature-responsive cis-elements, MeJA-responsiveness element. Class II contains regulatory elements involved in Me-JA and gibberellin acid responsive cis-elements. Only the Class III members have functions involved in differentiation of the palisade mesophyll cells. Class IV contains cis-elements associated with ABA and salicylic acid responsive cis-elements. Interesting, almost all *BpSWEET* genes have anaerobic induction elements. This result proved that different gene subfamily members may have functional differentiations.

### 2.5. Transcript Analysis of BpSWEETs in Different Organs/Tissues, Cold Stress

On the basis of published birch transcriptomic data, the transcript levels of *BpSWEET* genes in different organs and tissues were analyzed (Appendix A). All values were normalized for each row after log2 conversion, and the results of the expression level based on transcriptomic data was exhibited as Figure 6. As can be seen from the Figure 6, *BpSWEET* gene expressions were different among flowers, leaves, roots and xylem. Moreover, there were two genes (*BpSWEET4*, *BpSWEET9*) that have no expression data in any organs and tissues (the RNA seq data all showed 0). Among the other 11 *BpSWEET* genes, the gene with the highest expression levels in flowers, leaves and roots was *BpSWEET1c* (Figure 6A), and the gene with the highest expression in xylem was *BpSWEET1b* (Figure 6A). Eight genes were expressed in all four organs/tissues, while three genes were expressed in individual organs/tissues. In detail, *BpSWEET10* was only expressed in roots, *BpSWEET11* was expressed in flowers and roots, *BpSWEET15* was expressed in flowers, roots and xylem, but not in leaves (Figure 6A). The results suggest that different *BpSWEET* genes may play various roles in development of birch.

In order to reveal the roles of *BpSWEET* genes in the cold stress resistance of birch, the transcriptome data of *BpSWEET* genes in leaves under cold stress were analyzed based on published data (Appendix A) (Figure 6) [22]. All values were normalized for each row after log2 conversion and shown as a heatmap, similar to expression data of *BpSWEETs* in different tissues and organs; *BpSWEET4* and *BpSWEET9* have no expression data at the control or any of the six time points of cold stress treatment (the RNA seq data all showed 0). Most genes were up-regulated after 6 °C treated, while the time point of reaching the highest expression level was different. For example, *BpSWEET3* and *BpSWEET7* had the highest expression at 6 °C for 1 h, *BpSWEET2* and *BpSWEET17a* reached their highest expression level at 1.5 h, *BpSWEET1b* and *BpSWEET1c* reached the highest expression level at 3 h (Figure 6B). In general, the results showed that most genes in the *BpSWEET* gene family could rapidly change gene expression in response to cold stress.

In order to verify the RNA-seq data, seven *BpSWEET* genes (*BpSWEET1a*, *BpSWEET1b*, *BpSWEET1c*, *BpSWEET2*, *BpSWEET7*, *BpSWEET17a*, *BpSWEET17b*) responding to cold stress were selected for quantitative verification. The results showed that the expression trend of these seven genes after cold treatment was basically consistent with the results of RNA-seq data (Figure 7).

### 2.6. Functional Analysis of BpSWEET1c

The *BpSWEET1c*, which has the highest expressed peak of *BpSWEET* genes in cold stress and roots (Appendix A), was selected for further analysis. To verify the subcellular localization predicted by the WoLF PSORT, the BpSWEET1c-GFP fusion protein was transferred into tobacco leaves through the transient transformation method. The results showed that the GFP-only protein was detected in the plasma membrane and nucleus, etc, whereas BpSWEET1c was only localized in the plasma membrane (plas) (Figure 8), which was consistent with the subcellular localization results predicted by the website.

In previous studies, AtSWEET1, a homologue of BpSWEET1c, was involved in intercellular sugar transport as a glucose transporter. In order to identify the function of BpSWEET1c, the pBI121-*BpSWEET1c* vector was transferred into birch through the transient transformation method to obtain overexpressed plants. At the same time, the birch transformed with transient empty vector was treated as the control. After two days of co-culture, the relative expression of *BpSWEET1c* in the overexpressed plants was significantly 8 times higher than that in the control plants (Appendix A). We found that the expression of *BpSWEET1c* increased in both leaves and roots under cold stress (Figure 6, Figure 7 and Appendix A). Furthermore, the glucose, fructose, sucrose and Malondialdehyde (MDA) contents in roots and leaves were measured in both the *BpSWEET1c*-OE plants and the control (Figure 9). In previous studies, plants were usually subjected to cold stress at 4 °C. Compared with 6 °C, the physiological index changes of plants under 4 °C stress were more obvious [23], so we chose to stress the *BpSWEET1c*-OE plants and control at 4 °C for 3 h and 6 h.

The results showed that the glucose content of the OE lines was always lower than the control in the leaves, while it was gradually increased as the stress time increased in the roots. However, most of the results did not show significant differences. There was a significant difference only at 0 h in leaves, which indicated that *BpSWEET1c* may be involved in glucose transport, but it does not exert a protective effect against cold stress through glucose transport. Interesting, at 6 h of 4 °C in the leaves and at all studied time points in the roots, the sucrose content of the *BpSWEET1c*-OE plants was significantly higher than that of the control. It is speculated that *BpSWEET1c* may be involved in sucrose metabolism and transport, and may play a role through sucrose metabolism and transport during cold stress. In addition, there was no significant difference in fructose content between the *BpSWEET1c*-OE plants and the control in the leaves and roots.

MDA is the final decomposition product of membrane peroxidation, and MDA content can reflect plant resistance to cold [24]. The MDA content showed no significant difference between the OE and the control in the leaves. In roots, the MDA content in the control was significantly higher than that in the OE lines at 4 °C for 6 h, indicating that overexpression of *BpSWEET1c* enhances the resistance of birch to cold. The results suggest that *BpSWEET1c* may play a positive role in the response to cold stress by involving soluble sugars metabolized.

## 3. Discussion

An increasing number of species have been identified in *SWEET* family genes. For example, 17, 13, 17, 29, 21, 23, 16, 52 *SWEET* genes have been identified in Arabidopsis [9], tea [25], grape [26], tomato [27], rice [28], sorghum [29], litchi [30] and soybean [31]. *SWEET* genes of birch have not been reported in previous studies. A total of 14 candidate *SWEET* genes were identified in the birch genome; one gene was excluded because of an incomplete genetic structure. A total of 13 *BpSWEET* genes can be divided into four Clades. Clade I, II, III and IV contain 5, 2, 4 and 2 genes, respectively (Figure 1), and these results are also consistent with Chen’s classification of the *SWEET* gene family [9]. Each clades contained some unique elements (Figure 2), suggesting that genes in different clades may have different functions.

Recently, more and more studies have proved that the number of *SWEET* genes in plants is caused by tandem repetition events [32] and chromosome replication events [33]. The analysis of the collinearity of the *BpSWEET* family genes and their distribution on chromosomes showed that there is indeed a pair of tandem repeat genes (*BpSWEET1b/1c*) and two pairs of chromosome replication genes (*BpSWEET11/15*, *BpSWEET17a/17b*) in the *BpSWEET* gene family (Figure 3). Interestingly, *BpSWEET1a*, *BpSWEET1b*, and *BpSWEET1c* all belong to Clade I and have motif 6, which is located very close on the Chr02. However, MCScanX analysis results showed that *BpSWEET1a* is not a tandem repeat gene of *BpSWEET1b*, *BpSWEET1c*. The results indicate that *BpSWEET1a* may have undergone evolutionary differentiation and is worthy of further research.

In previous studies, the *SWEET* genes had different expression patterns in various organs and tissues, thus performing different functions [25,34]. In this study, *BpSWEET* genes were highly expressed in flowers and roots but were less expressed in leaves (Figure 6). In addition, the duplicated genes in birch also had different expression patterns in different organs and tissues. For example, the tandem repeat genes, *BpSWEET1b/1c*, have different expression patterns. *BpSWEET1b* is highly expressed in xylem, while *BpSWEET1c* is highly expressed in roots. Our results were similar to the expression patterns of tandem repeat genes in litchi [30], apple [35] and wheat [36], suggesting the functional divergence of *SWEET* duplicated genes. *BpSWEET11/15* genes derived from chromosomal replication events exhibited similar expression patterns, with high expression in both flowers and roots, similar to the expression pattern of *PagSWEET15* in *P. trichocarpa* [37]. *BpSWEET17a* and *BpSWEET17b* are highly expressed in flowers, while *AtSWEET16* and *AtSWEET17* are mainly expressed in roots and leaves [38], which may be caused by the difference between birch and Arabidopsis. In contrast, *PagSWEET17b*, *PagSWEET17c* and *PagSWEET17d* in *P. trichocarpa* have higher expression in flowers [37], which verifies the hypothesis that the relationship between birch and *P. trichocarpa* is closer through collinearity analysis.

Most *BpSWEET* gene promoters have cis-acting elements in response to stress, especially in response to low temperature stress. Compared to normal conditions, the relative expression levels of most *BpSWEET* genes were up-regulated within 3 h after cold stress (Figure 6), indicating that the *BpSWEET* gene family can rapidly participate in response to cold stress. In the process of plant evolution, tandem repeat events lead to the generation of a series of tandem repeat genes, which play a very important role in the process of plant adaptation to environmental changes [39]. In Arabidopsis and rice genomes, tandem repeat genes play important roles in enhancing stress tolerance and membrane function [40]. In addition, there is many abiotic stress-related genes in the genome of *S. thaliana*, which are mainly produced by tandem repetition, making *S. thaliana* more tolerant to salt and cold [41]. In the *BpSWEET* gene family, the expression of *BpSWEET1b/1c* highly increased under cold stress. In particular, *BpSWEET1c* has the highest expression peak in *BpSWEET* genes after 3 h cold treatment, suggesting that *BpSWEET1c* plays a key role in resisting cold stress, which is worthy of further study. In addition, as a tandem repeat gene of *BpSWEET1c*, the expression of *BpSWEET1b* also changes significantly under cold stress, which is worth further research in future work.

The accumulation of sugars in cells can improve the cell’s tolerance to cold. As an osmoprotectant and antioxidant, sugar can stabilize cell membranes and protect the integrity of membranes by interacting with membrane lipids [42]. In addition, sugars can protect phospholipids in the cell membrane system by inducing the formation of glass in the cytoplasm [43]. In this study, the glucose content was measured and it was found that the glucose content of overexpressed *BpSWEET1c* lines in leaves was significantly higher than that of the control, but there was no significant difference in roots and leaves after cold stress. We speculate that BpSWEET1c proteins may be involved in glucose metabolism and transportation, but it does not play a significant role in resisting cold stress, suggesting that BpSWEET1c may also affect other soluble sugars to resist cold stress.

The *CsSWEET1a*-OE Arabidopsis lines had higher sucrose content than the wild-type, suggesting that CsSWEET1a may affect sucrose metabolism and transport [44]. Further analysis showed that *BpSWEET1c*, *CsSWEET1a* all had seven Transmembrane domains in the amino acid sequence. Under cold stress, *BpSWEET1c* and *CsSWEET1a* genes were up-regulated, which could respond to cold stress. In this study, *BpSWEET1c*-OE lines also had higher sucrose content than the control at 6 h of 4 °C in the leaves and all studied time points in the roots, and the sucrose content of both groups increased gradually with the increase of stress time (Figure 9), indicating that BpSWEET1c proteins may also affect sucrose metabolism and transport and play a positive role in the response to cold stress by increasing the sucrose content in plants. Interestingly, the fructose content of *BpSWEET1c*-OE in roots decreased gradually with stress time (Figure 9). Similar results were also found in the *AtSWEET16*-OE Arabidopsis lines [38] and the *CsSWEET16*-OE Arabidopsis lines [25]. There are seven Transmembrane domains in the amino acid sequences of *BpSWEET1c*, *AtSWEET16* and *CsSWEET16*. Although the expression levels of *AtSWEET16* and *CsSWEET16* are down-regulated under cold stress, *BpSWEET1c*, *AtSWEET16* and *CsSWEET16* can respond to cold stress significantly. We guess that sucrose is more effective in protecting plants from low-temperature damage than fructose. However, if external factors stimulate BpSWEET1c proteins to undergo post-translational modification, the impact on sugar transport activity cannot be included. In conclusion, we demonstrated that *BpSWEET1c* can respond to cold stress by affecting soluble sugars’ metabolisms using the transient transformation method. However, the specific functions of *SWEET* genes in birch have not been determined in this study and need to be further explored.

## 4. Materials and Methods

### 4.1. Identification of SWEET Family Genes in Birch and the Acquisition of Genomic Information for Other Species

To identify *SWEET* gene family members in birch, 17 known *SWEET* gene family members in Arabidopsis were searched in TAIR (https://www.arabidopsis.org/, accessed on 1 February 2023). The Arabidopsis SWEET proteins sequences were used as query to search the *SWEET* genes in birch genome [17] (https://phytozome-next.jgi.doe.gov/info/Bplatyphylla_v1_1, accessed on 1 February 2023). Fourteen *BpSWEET* gene family members were identified using the BLASP method, and one gene has been excluded due to the incomplete gene structure. The conserved domains were found in the NCBI Conserved Domain Database for the protein sequences of all identified genes [45,46]. Gene sequences without MtN3_slv or PQ-loop superfamily domains were removed. In addition, the genome data of *O. sativa*, *P. trichocarpa*, *V. vinifera*, *A. trichopoda* and *S. lycopersicum* was downloaded from the phytozome13 (http://phytozome-next.jgi.doe.gov, accessed on 1 February 2023).

### 4.2. Analysis of Main Characteristics and the Phylogenetic Tree Analysis of SWEET Genes in Birch

Protein length, molecular weight, isopotential point and other information were analyzed using the ProtParam tool on Expasy (http://us.expasy.org/tools/protpara-m.html, accessed on 1 February 2023). The SWEET protein transmembrane domains were predicted by DeepTMHMM (http://DTU/DeepTMHMM-BioLib, accessed on 1 February 2023). Subcellular localization of *SWEET* genes in birch was achieved through the WoLF PSORT (http://wolfsort.hgc.jp, accessed on 1 February 2023). In addition, TBtools [47] was used to search the 2000 bp upstream regions of the start codon of the *BpSWEET* genes. Plantcare (http://bioinformatics.psb.ugent.be/webtools/plantcare/html, accessed on 1 February 2023) were used to identify cis-elements in the promoter regions.

MUSCLE was used to compare the SWEETs protein sequences of Arabidopsis and birch, and the protein sequences of *BpSWEET* family members alone [48]. The results of multiple sequence alignment were presented by GeneDoc. The phylogenetic tree was constructed by Neighbor-Joining (NJ) method on MEGA11.0, the bootstrap value was set to 1000. The phylogenetic tree results were presented with Evolview.

### 4.3. Gene Structures and Conserved Protein Motifs Analysis of BpSWEET Genes

The gene structure information of the *BpSWEET* genes, including introns and exons, was obtained in the birch genome annotation file and presented by TBtools [47]. MEME (http://meme-suit.org/meme/tools/meme, accessed on 1 February 2023) was used to analyze the motifs in BpSWEET protein sequences, the number of motifs searched was set to 10, and the site distribution was set to Zero or One Occurrence Per Sequence (zoops).

### 4.4. Identification and Analysis of Orthologous Gene and Paralogous Gene of SWEET Genes

The orthologous gene identification of *SWEET* genes in birch and the paralogous gene identification of birch with six other species were achieved by TBtools [47]. The collinear analysis of these homologous genes was further performed and TBtools was used for mapping. The synonymous mutation rates (ks) and non-synonymous mutation rates (ka) of *BpSWEET* gene pairs were calculated by TBtools [47].

### 4.5. Expression Analysis of BpSWEET Genes and Verification of RNA-Seq Data by qRT-PCR

To analyze the expression of *BpSWEET* genes in different organs/tissues, the raw data of organs and tissues (leaves, roots, flowers and xylem) from birch were obtained from NCBI (accession code: PRJNA 535361). To analyze the expression of *BpSWEET* genes under cold stress, the transcription data of birch under the cold treatment were obtained from NCBI (accession code: PRJNA532995). The TPM of the *BpSWEET* genes was used to construct the heat map. Heat maps of *BpSWEETs* in different organs and tissues and under cold stress was presented by TBtools. Using real time fluorescence quantitative gene amplification instrument (qTOWER3G), the qRT-PCR experiment was performed. Some gene expression data of leaves RNA-Seq were further identified by qRT-PCR. The corresponding primers were designed according to the coding region of *BpSWEET* genes (Appendix A).

### 4.6. Subcellular Localization Analysis of BpSWEET1c Protein

The corresponding primers were designed according to the coding region of the *BpSWEET1c* gene (Appendix A). The full-length sequence of the target gene is amplified with forward and reverse primers and connected to the pBI-121 vector. The recombinant vector is transformed into Agrobacterium GV3101, the infiltration and transient expression of tobacco is performed using the previously published protocols [49]. Two days after agrobacterium infection, the results were observed by confocal laser scanning microscope (Zeiss, Jena, Germany, LSM 800) and the GFP fluorescence photographs were obtained.

### 4.7. Transient Transformation and Related Physiological Indexes of Birch

The method of transient transformation referred to the method of Dong et al. [50]. Specifically, Agrobacterium Tumefaciens containing the target vector was cultured in LB medium containing the screening agent at 28 °C until OD_600_ reached 0.6–0.8. The culture medium was centrifuged at 3000× *g* with a high-speed centrifuge, and the obtained bacteria were suspended in the transformation solution [1/2 MS (Murashige and Skoog medium) + 2.5% (*w*/*v*) sucrose + 120 μM acetosyringone + Tween (0.02%, *v*/*v*), pH 5.4] until OD_600_ reached 0.7–0.8. Birch was soaked in the transformation solution at 25 °C for 2 h with shaking conditions for infection. After washing the transformation solution on birch with sterile distilled water, the birch was planted on solid MS medium [MS + 120 μM acetosyringone + 2% (*w*/*v*) sucrose + 0.5 mg/L NAA + 2 mg/L 6-BA, pH 5.8] for 48 h before the next experiments. The expression of *BpSWEET1c* in transient overexpressed *BpSWEET1c* gene birch (OE) was detected by qRT-PCR; the corresponding primers were designed according to the coding region of the *BpSWEET1c* gene (Appendix A).

An amount of 0.05 g~0.1 g plant material was used to measure. MDA follows the method of Liu et al. [51]. The glucose and sucrose content were determined using a corresponding sugar content determination kit (Nanjing Jiancheng Bioengineering Research Institute, Nanjing, China). Fructose was measured using a fructose content measurement kit (Suzhou Keming Biotechnology Co., Ltd., Suzhou, China).

### 4.8. Statistical Analyses

Student’s *t*-test and one-way analysis of variance was used for statistical analysis; * indicates a significant difference (*p* < 0.05), ** represents a very significant difference (*p* < 0.01).

## 5. Conclusions

In this study, 13 *BpSWEET* genes were identified from the birch genome, and gene structures, chromosome localization, collinearity and cis-acting elements of the promoter regions were analyzed. Through RNA-seq data, we found that *BpSWEET* genes were highly expressed in several sink organs, and rapidly changed gene expression in response to cold stress. Moreover, *BpSWEET1c* was found to be located in the cell membrane, and transient overexpressed *BpSWEET1c* may play a positive role in the response to cold stress by changing the soluble sugar content. The identification and preliminary analysis of the *SWEET* gene family will lay a good foundation for future research on the relationship between sugar and abiotic stress.

## Figures and Tables

**Figure 1 ijms-24-13626-f001:**
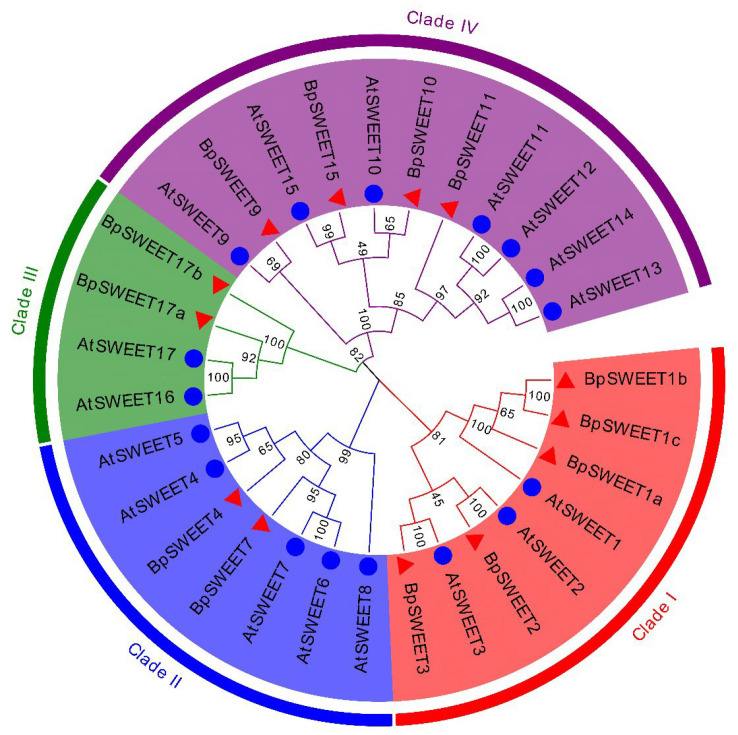
Analysis of the phylogenetic relationship between the SWEET proteins in birch and Arabidopsis. A total of 17 AtSWEEP proteins and 13 BpSWEET proteins conserved domains were aligned through MUSCLE. An evolutionary tree was constructed using Neighbor-Joining (NJ) method at MEGA11.0, the bootstrap value was set to 1000, Clades I, II, III and IV are represented by red, blue, green and purple, respectively. The evolutionary tree was visualized through the online tool Evolview.

**Figure 2 ijms-24-13626-f002:**
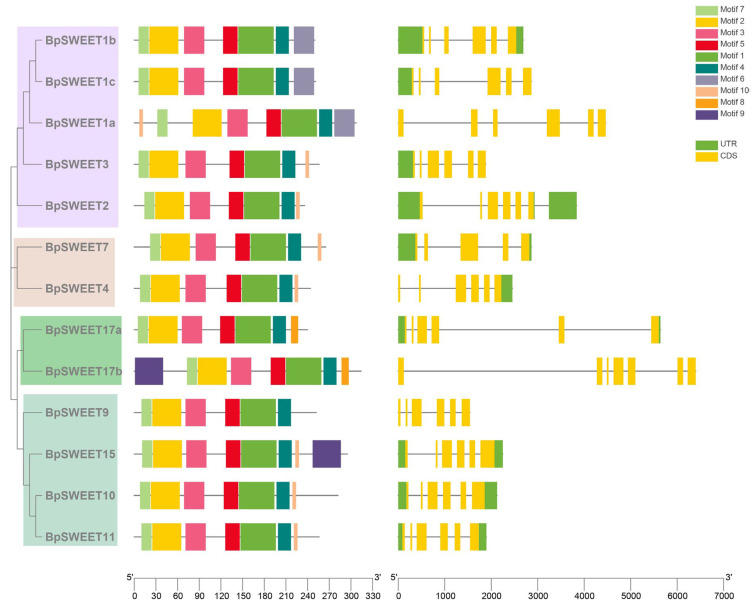
The conserved domain motifs and gene structures of 13 *BpSWEETs*. (**a**) Phylogenetic tree of 13 *BpSWEETs*. (**b**) Distribution of motifs on SWEET proteins. Motifs were analyzed by MEME, and a total of 10 different motifs were identified. Motifs 1–10 were presented with different colors. TBtools were used for visualization. (**c**) The genetic structures of the *BpSWEET* genes, with yellow boxes representing exons and black lines representing introns.

**Figure 3 ijms-24-13626-f003:**
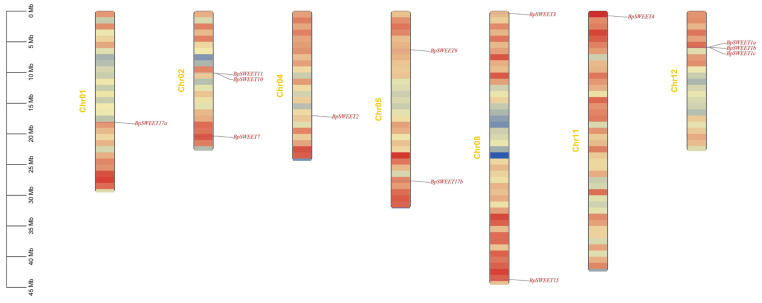
The distribution of 13 *BpSWEET* genes on the chromosomes of birch, visualized through TBtools.

**Figure 4 ijms-24-13626-f004:**
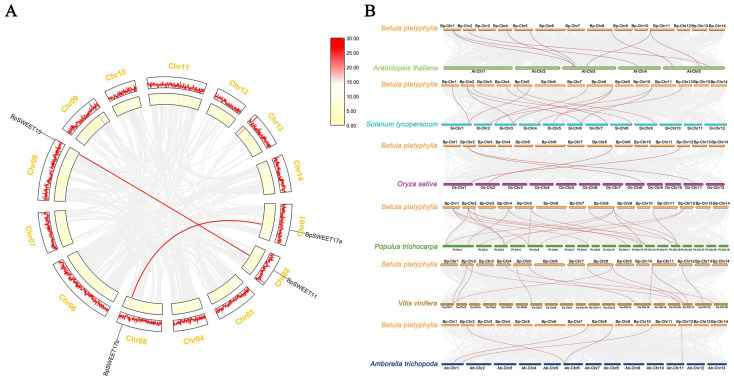
Collinearity of the *BpSWEETs*. (**A**) The relationship between *BpSWEETs* within the chromosomes of birch was determined through multiple collinear scanning toolkits (oneStepMCScanX-superFast in TBtools 2.0) and visualized through TBtools. The gray line indicates the collinearity modules within the birch genome, and the red line indicates the *SWEET* collinearity genes. (**B**) The collinearity analysis of *SWEET* genes in birch and six other species; the gray line represents the collinear blocks in the genome of birch and six other species, and the red line represents the *SWEET* collinearity genes in birch and six other species.

**Figure 5 ijms-24-13626-f005:**
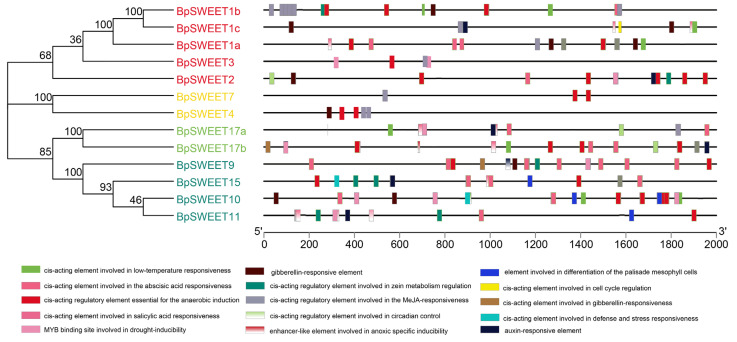
Analysis of cis-acting elements in the 13 *BpSWEETs* promoter regions; different colored boxes represent different cis-acting elements, and black lines represent the promoter length of the *BpSWEET* genes.

**Figure 6 ijms-24-13626-f006:**
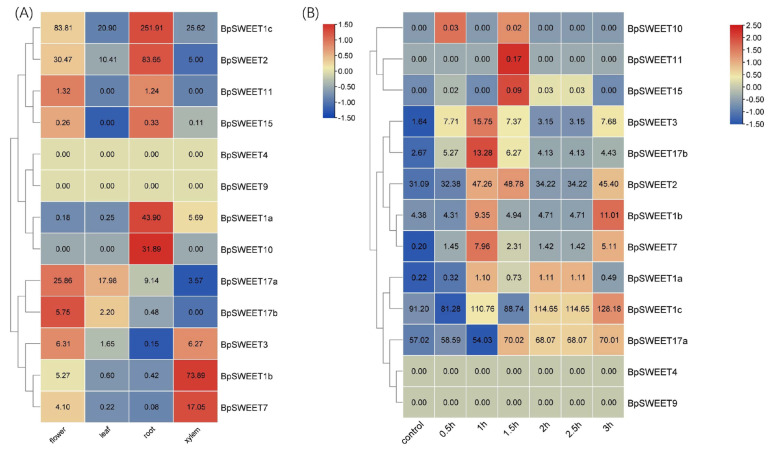
*BpSWEET* genes expression pattern based on RNA-seq analysis. Transcriptome data were obtained from NCBI (accession code: PRJNA535361 and PRJNA532995). Gene expression levels were shown as RNA-seq values log2 transformed followed by normalization for each row (**A**) Heat map of *BpSWEET* gene expression pattern in flowers, leaves, roots and xylem of two-month-old birch. (**B**) Heat map of *BpSWEET* gene expression pattern of two-month-old birch treated 6 °C for 0.5, 1, 1.5, 2, 2.5, and 3 h.

**Figure 7 ijms-24-13626-f007:**
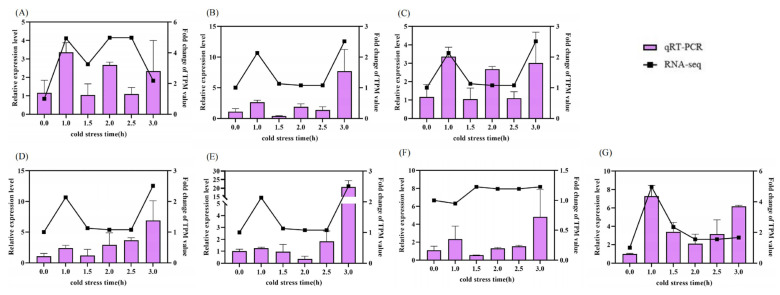
The expression of *BpSWEET* genes was verified by qRT-PCR. The bar graph is the relative expression level of qRT-PCR, and the line graph is the fold change of TPM value in RNA-seq data. The left *Y*-axis represents the numerical scale in the bar graph, and the right *Y*-axis represents the numerical scale in the line graph. (**A**) Changes in qRT-PCR and RNA-seq values of *BpSWEET1a*. (**B**) Changes in qRT-PCR and RNA-seq values of *BpSWEET1b*. (**C**) Changes in qRT-PCR and RNA-seq values of *BpSWEET1c*. (**D**) Changes in qRT-PCR and RNA-seq values of *BpSWEET2*. I Changes in qRT-PCR and RNA-seq values of *BpSWEET7*. (**F**) Changes in qRT-PCR and RNA-seq values of *BpSWEET17a*. (**G**) Changes in qRT-PCR and RNA-seq values of *BpSWEET17b*.

**Figure 8 ijms-24-13626-f008:**
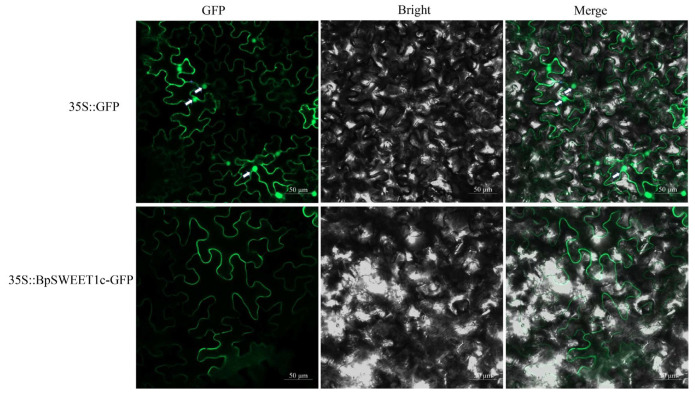
The subcellular localization of *BpSWEET1c* in tobacco leaves. *BpSWEET1c*-GFP and individual GFP empty vectors were instantly transformed into tobacco leaves through Agrobacterium infection. The experimental results are observed two days after infection. The white arrow shows where the nucleus is located.

**Figure 9 ijms-24-13626-f009:**
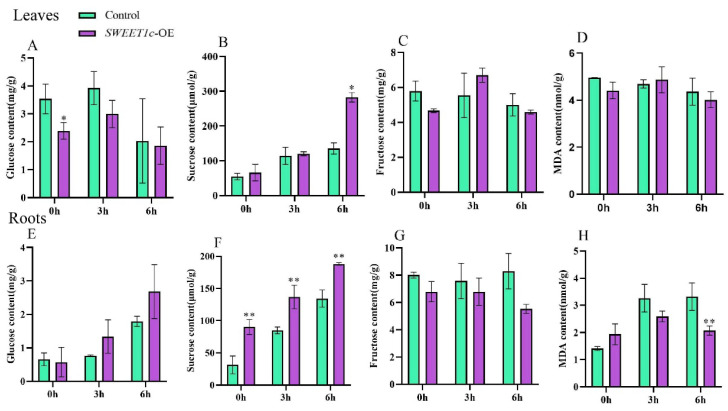
Analysis of glucose, sucrose, fructose and MDA contents of *BpSWEET1c*-OE and pBI-121 birch lines at 4 °C for 0, 3 and 6 h in leaves and roots. (**A**) Change of glucose content in leaves. (**B**) Changes of sucrose content in leaves. (**C**) Changes of fructose content in leaves. (**D**) Changes of MDA content in Ives. (**E**) Changes of glucose content in roots. (**F**) Change of sucrose content in roots. (**G**) Changes of fructose content in roots. (**H**) Changes of MDA content in roots. * Represents a significant difference (*p* < 0.05). ** Represents a very significant difference (*p* < 0.01).

**Table 1 ijms-24-13626-t001:** SWEET gene information for birch.

Gene ID	Gene Name	GenBank Accession Numbers	Protein Length (aa)	MW (kD)	pl	POSRT Predictions	Number of MtN3_slv/PQ-Loop Repeat Superfamily Domain	Transmembrane Domain
BPChr12G08384.v1.1.679	*BpSWEET1a*	OR361584	306	33.98	8.9	plas	2	7
BPChr12G08398.v1.1.679	*BpSWEET1b*	OR361585	249	27.49	9.54	plas	2	7
BPChr12G08357.v1.1.679	*BpSWEET1c*	OR361586	250	27.66	9.46	plas	2	7
BPChr04G06173.v1.1.679	*BpSWEET2*	OR361587	235	26.18	8.25	plas	2	7
BPChr08G24178.v1.1.679	*BpSWEET3*	OR361588	255	28.53	9.25	plas	2	7
BPChr11G26889.v1.1.679	*BpSWEET4*	OR361589	243	26.98	9.46	plas	2	7
BPChr02G10545.v1.1.679	*BpSWEET7*	OR361590	264	29.39	8.96	plas	2	7
BPChr05G07871.v1.1.679	*BpSWEET9*	OR361591	251	28.29	8.26	vacu	2	7
BPChr02G23316.v1.1.679	*BpSWEET10*	OR361592	281	31.91	9.09	plas	2	7
BPChr02G23301.v1.1.679	*BpSWEET11*	OR361593	255	28.54	9.18	plas	2	7
BPChr08G16703.v1.1.679	*BpSWEET15*	OR361594	294	33.22	5.79	plas	2	7
BPChr01G13604.v1.1.679	*BpSWEET17a*	OR361595	239	26.63	8.94	vacu	2	7
BPChr05G22191.v1.1.679	*BpSWEET17b*	OR361596	313	35.49	8.7	plas	2	7

Note: MW: molecular weight; pl: isoeletric points; plas: plasma membrane; vacu: vacular membrane.

## Data Availability

Not applicable.

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
