# Peer review of "An Insight of Betula platyphylla SWEET Gene Family through Genome-Wide Identification, Expression Profiling and Function Analysis of BpSWEET1c under Cold Stress"

_ijms, 2023, doi:10.3390/ijms241713626_

Round 1

Reviewer 1 Report

English language need to be improved.

Author Response

Dear Editor:

Thank you for your good work on our manuscript entitled “An insight of Betula platyphylla SWEET gene family through Genome-wide identification, expression profiling and function analysis of BpSWEET1c under cold stress” (Manuscript ID: ijms-2503004), and we are very grateful to the reviewer’s valuable comments and suggestions. We carefully answered all the reviewers' questions and pointed out the location of the revisions, which raised the level of our manuscript and made it more in line with the requirements and standards of the journal. Relevant changes are highlighted in blue font in the revised manuscript If there are further issues to be clarified, please contact us without hesitation.

Best regards,

Caiqiu Gao

State Key Laboratory of Tree Genetics and Breeding (Northeast Forestry University),

26 Hexing Road, Harbin 150040, China

Tel: +86-451-82191820

Response to Reviewer 1 Comments

Point 1: Line 157, BpSWEET1c in OE line. What is meaning of OE line? Nothing has been mentioned about it.

Responses 1: Thanks for your suggestion, OE line is meaning the transient overexpressed BpSWEET1c gene birch. We have added it in the manuscript (line160-161 in the revised manuscript).

Point 2: Please deposit BpSWEET gene sequences in NCBI and mention gene accession number of all 14 BpSWEET in table 1?

Responses: Thanks for your suggestion, we have submitted BpSWEETs and related information at NCBI, the specific GenBank accession numbers is OR361584- OR361596. And the related information has added in table 1 in the revised manuscript. 

Point 3: Discuss multiple sequence alignment of 14 BpSWEET in manuscript.

Responses: Thanks for your suggestion, we have performed multiple sequence alignment of BpSWEETs and explained it in the article 3.1 section (line110-112, 178-179 in the revised manuscript).

Point 4: Please validate BpSWEET gene expression by qRT PCR.

Responses: Thanks for your suggestion, seven BpSWEET genes (BpSWEET1a, BpSWEET1b, BpSWEET1c, BpSWEET2, BpSWEET7, BpSWEET17a, BpSWEET17b) responding to cold stress were selected for quantitative verification. Related content has been added in 3.5 and Figure7 (line135-139, line319-323, Figure7 in the revised manuscript).

Reviewer 2 Report

Authors pretended to re-identify sweet gene family in Betula platyphylla. Unfortunately the genome of Betula platyphylla was published and all genes already were professionally annotated. A quick search in the database easily results in 14 sweet genes, with all relevant information including exon structure, sequences and description. So the current paper does not present any new result up to section 3.3.

Promoter analysis is a complete mess. the presented motifs cannot be arranged specifically to any of the groups of genes.”A large part of the Class I family contains hormones-responsive and 

low temperature-responsive cis-elements.” - This statement cannot be concluded from the figure 5 and therefore is a misinterpretation of the results. 

Section 3.6. From the figure I see that the gfp in both cases is distributed on the membrane therefore I see no difference in distribution of the protein. Authors should better explain their experiment and evidently show their results.

“the BpSWEET1c under low temperature stress has the highest relative expression” - This is the false statement as follows from the Figure 6 - for example gene sweet11 has a higher expression.

 “These results indicating that BpSWEET1c may improve low temperature tolerance” - How authors show that there is no indication on a low temperature tolerance or susceptibility for BpSWEET genes.

In summary, authors strongly misinterpret the data they downloaded from different sources. in most cases the conclusions cannot be justified with data presented on  figures. 

English of the paper must be significantly improved following are several examples 

-An increasing number of species have identified SWEET family genes in botany

- conserved motifs about 13 BpSWEET genes

 - Gene structures diversity  - what is that?

- may be conservative in evolutionary.

Author Response

Dear Editor:

Thank you for your good work on our manuscript entitled “An insight of Betula platyphylla SWEET gene family through Genome-wide identification, expression profiling and function analysis of BpSWEET1c under cold stress” (Manuscript ID: ijms-2503004), and we are very grateful to the reviewer’s valuable comments and suggestions. We carefully answered all the reviewers' questions and pointed out the location of the revisions, which raised the level of our manuscript and made it more in line with the requirements and standards of the journal. Relevant changes are highlighted in blue font in the revised manuscript If there are further issues to be clarified, please contact us without hesitation.

Best regards,

Caiqiu Gao

State Key Laboratory of Tree Genetics and Breeding (Northeast Forestry University),

26 Hexing Road, Harbin 150040, China

Tel: +86-451-82191820

Response to Reviewer 2 Comments

Point 1: Authors pretended to re-identify sweet gene family in Betula platyphylla. Unfortunately the genome of Betula platyphylla was published and all genes already were professionally annotated. A quick search in the database easily results in 14 sweet genes, with all relevant information including exon structure, sequences and description. So the current paper does not present any new result up to section 3.3.

Responses: Thanks for the suggestion. SWEET proteins, as important sugar transporter proteins, have very important value in plant research. Although 14 BpSWEET genes can be found in Betula platyphylla genome, the physicochemical properties, evolutionary relationships and conserved domain information of the sequences have not been reported. We identified its gene family and analyzed its genetic relationship with AtSWEETs by constructing an evolutionary tree (Figure1). According to the traditional naming method of the SWEETs, the SWEETs in birch was named based on its genetic relationship with the AtSWEETs. Through chromosome localization and collinear analysis (Figure3, Figure4), one group tandem duplication gene pairs and two pairs of segmental duplication genes were found. In addition, through motif analysis showed that different BpSWEET genes had different motif, which might be related to their functions (line211-216 in the revised manuscript). These results laid a good foundation for the subsequent study of BpSWEET genes.

Point 2: Promoter analysis is a complete mess. The presented motifs cannot be arranged specifically to any of the groups of genes. “A large part of the Class I family contains hormones-responsive and low temperature-responsive cis-elements.” This statement cannot be concluded from the figure 5 and therefore is a misinterpretation of the results.

Response Thank you for your suggestion. We further simplify the number of Cis-regulatory elements in the promoter region for easy observation, and further explained in the results section (line280-285 in the revised manuscript).

Point 3: Section 3.6. From the figure I see that the gfp in both cases is distributed on the membrane therefore I see no difference in distribution of the protein. Authors should better explain their experiment and evidently show their results.

Response: In Figure 7, the GFP protein is located in the membrane and nucleus, while the BpSWEET1c-GFP protein is located on the membrane. To show this more clearly, we add an indication of the nucleus and a clearer description in the manuscript (line345-346 in the revised manuscript, Figure8).

Point 4: “the BpSWEET1c under low temperature stress has the highest relative expression” - This is the false statement as follows from the Figure 6 - for example gene sweet11 has a higher expression.

Response: We are very sorry for the incorrect display. In the RNA seq data, there is a significant difference in numerical values. In order to demonstrate the variation pattern, gene expression levels were shown as RNA-seq values log2 transformed followed by row normalization for each row, and the heat map was displayed. To avoid any confusion, we illustrated this in the diagram notes (line326-327 in the revised manuscript) and show the original TPM values in the heat map (Figure6). In addition, we added the RNA-seq raw data in the supplementary material (TableS4, S5).

Point 5: “These results indicating that BpSWEET1c may improve low temperature tolerance” - How authors show that there is no indication on a low temperature tolerance or susceptibility for BpSWEET genes.

Response: Thank you for your suggestion. We think it is more appropriate to state that BpSWEET1c may play a positive role in the response to cold stress. The relevant statements have been corrected in the article (line21-22 in the revised manuscript).

Point 6: English of the paper must be significantly improved following are several examples.

-An increasing number of species have identified SWEET family genes in botany

- conserved motifs about 13 BpSWEET genes

- Gene structures diversity - what is that?

- may be conservative in evolutionary.

Responses: Thank you for your guidance. We have reviewed and made changes to the manuscript. The relevant errors have been corrected in the article.

Reviewer 3 Report

Manuscript An insight of Betula platyphylla SWEET gene family through

Genome-wide identification, expression profiling and function

analysis of BpSWEET1c under low temperature stress

Hao Zhang et al., investigates the effect of the soluble sugar content and improved the low temperature tolerance of birch.

The work is framed according to the rules of the journal, contains the necessary sections.

However, questions arise regarding various sections of the manuscript. The authors apparently forgot the need for the correct use of terms. In materials and methods, starting from line 129 and further in the text, the term tissue is mentioned.

This is a very specific concept. If xylem can be called tissue, then the rest contain many different tissues. For example, a leaf contains epidermis, mesophyll (spongy and columnar parenchyma, phloem, xylem, meristematic tissues - protoxylem and protophloem, often trichomes and stomata are isolated separately from the epidermal tissue). It’s the same and even worse if you call a flower a fabric. Not only is a flower a complex structure with different types of tissues, but it can also contain several genomes of different ploidy and structure.

The question of how exactly the authors isolated xylem also remains mysterious.

I think it is worth working on the materials and methods section so that this material does not become an embarrassment for the authors and the journal.

It is also worth carefully straightening the text, taking into account this incident of confusion.

each tissue (leaf, root, flower, and xylem)???

There are separate questions regarding the application of the cold stress to temperatures of 4-6 C. All the same, stress is also a term. In this case, the authors worked with low positive temperatures, and this is not the same thing. Be careful and discuss it.

Author Response

Dear Editor:

Thank you for your good work on our manuscript entitled “An insight of Betula platyphylla SWEET gene family through Genome-wide identification, expression profiling and function analysis of BpSWEET1c under cold stress” (Manuscript ID: ijms-2503004), and we are very grateful to the reviewer’s valuable comments and suggestions. We carefully answered all the reviewers' questions and pointed out the location of the revisions, which raised the level of our manuscript and made it more in line with the requirements and standards of the journal. Relevant changes are highlighted in blue font in the revised manuscript If there are further issues to be clarified, please contact us without hesitation.

Best regards,

Caiqiu Gao

State Key Laboratory of Tree Genetics and Breeding (Northeast Forestry University),

26 Hexing Road, Harbin 150040, China

Tel: +86-451-82191820

Response to Reviewer 3 Comments

Point 1: The authors apparently forgot the need for the correct use of terms. In materials and methods, starting from line 129 and further in the text, the term tissue is mentioned. This is a very specific concept. If xylem can be called tissue, then the rest contain many different tissues. For example, a leaf contains epidermis, mesophyll (spongy and columnar parenchyma, phloem, xylem, meristematic tissues - protoxylem and protophloem, often trichomes and stomata are isolated separately from the epidermal tissue). It’s the same and even worse if you call a flower a fabric. Not only is a flower a complex structure with different types of tissues, but it can also contain several genomes of different ploidy and structure.

Responses: Thank you for your guidance, all parts have been corrected in the article (line129-130,294-303 in the revised manuscript).

Point 2: The question of how exactly the authors isolated xylem also remains mysterious.

Responses: Thank you for your guidance. The data of RNA-seq in xylem was obtained from NCBI (accession code: PRJNA535361), and related papers were also published.

Chen, S., Wang, Y.C., Yu, L.L., Zheng, T., Wang, S., Yue, Z., et al. (2021). Genome sequence and evolution of Betula platyphylla. Horticulture Research 8(1). doi: 10.1038/s41438-021-00481-7.

Point 3: I think it is worth working on the materials and methods section so that this material does not become an embarrassment for the authors and the journal.

It is also worth carefully straightening the text, taking into account this incident of confusion.

Response: Thanks for your suggestions, we have reviewed the materials and methods section as well as the full text and revised the inappropriate places.

Point 4: There are separate questions regarding the application of the cold stress to temperatures of 4-6 C. All the same, stress is also a term. In this case, the authors worked with low positive temperatures, and this is not the same thing. Be careful and discuss it.

Responses: Thanks for your suggestions, we have replaced the term 'low temperature stress' in the text with a more rigorous term 'cold stress'. In previous studies, plants were usually subjected to low temperature stress at 4°C, so we chose to stress OE-line and control at 4°C. In addition, compared with 6°C, the physiological index changes of plants under 4°C stress were more obvious (Wu and Liu, 2007). In order to obtain more obvious physiological index changes, the instantaneous transformed birch was treated with 4°C (line362-365 in the revised manuscript).

Wu, Y., and Liu, J. (2007). Effects of chilling stress on chill-resistance physiological and biochemical indexes of muskmelon seedlings. Journal of northwest sci-tech university of agriculture and forestry 35(3), 139-143.

Reviewer 4 Report

The authors have found 13 sugar transporter SWEET genes from the genome sequence of Birch (Betula platyphylla) and analyzed their gene structure (Figures 1 to 4). Some had hormone- or cold-responsive cis-elements, like SWEET genes from other plants. Some of the BpSWEET genes were upregulated by cold stress (Figure 5). Of these, the authors focused on BpSWEET1c for functional analysis. The BpSWEET1c OE mutant (Figure S1), which expression level was 8-fold up-regulated compared to the wild type, resulted in higher sucrose content in roots and improved its cold tolerance (Figure 8). Although this manuscript does not contain truly novel findings and its scientific significance is not high, it may be useful as additional data collection in the related fields (e.g., Birch cold tolerance research, SWEET protein research, etc.).

Major comments

The following two points should be addressed to support the authors' claims.

1.     BpSWEET1c OE mutant showed an increase in sucrose content and cold tolerance in roots (Figure 8). To see if BpSWEET1c is involved in cold tolerance in the roots, the authors should analyze the expression changes of the BpSWEET1c gene in the roots of wild-type plants when subjected to cold stress.

2.     It has been known that SWEET proteins are involved in cold tolerance. The authors should explain it in the Introduction with citations of some papers; in the Discussion section, the authors should discuss the similarities and differences (of amino acid sequence, cis region sequence, expression pattern, etc.) between other plant cold tolerance-related SWEET proteins (CsSWEET1a, AtSWEET16, and many others) and BpSWEET1c.

Minor comment

3. Is the RNA analyzed in Figure 6B root-derived or leaf-derived?

Author Response

Dear Editor:

Thank you for your good work on our manuscript entitled “An insight of Betula platyphylla SWEET gene family through Genome-wide identification, expression profiling and function analysis of BpSWEET1c under cold stress” (Manuscript ID: ijms-2503004), and we are very grateful to the reviewer’s valuable comments and suggestions. We carefully answered all the reviewers' questions and pointed out the location of the revisions, which raised the level of our manuscript and made it more in line with the requirements and standards of the journal. Relevant changes are highlighted in blue font in the revised manuscript If there are further issues to be clarified, please contact us without hesitation.

Best regards,

Caiqiu Gao

State Key Laboratory of Tree Genetics and Breeding (Northeast Forestry University),

26 Hexing Road, Harbin 150040, China

Tel: +86-451-82191820

Response to Reviewer 4 Comments

Point 1: BpSWEET1c OE mutant showed an increase in sucrose content and cold tolerance in roots (Figure 8). To see if BpSWEET1c is involved in cold tolerance in the roots, the authors should analyze the expression changes of the BpSWEET1c gene in the roots of wild-type plants when subjected to cold stress.

Responses: Thanks for your guidance, the expression level of BpSWEET1c in the roots of wild-type plants under cold stress was analyzed. The relevant results are presented in Figure S3. In the Figure S3, it can be seen that BpSWEET1c can respond to low temperature stress in roots, and reached the highest level at 4°C 2.5 h (line359-360 in the revised manuscript).

Point 2: It has been known that SWEET proteins are involved in cold tolerance. The authors should explain it in the Introduction with citations of some papers; in the Discussion section, the authors should discuss the similarities and differences (of amino acid sequence, cis region sequence, expression pattern, etc.) between other plant cold tolerance-related SWEET proteins (CsSWEET1a, AtSWEET16, and many others) and BpSWEET1c.

Response: Thanks for your suggestion, More and more evidence suggest that the SWEET gene is involved in resisting cold stress. In the introduction, we cited some papers to explain this viewpoint (line55-66 in the revised manuscript).

In the discussion section, we discussed and analyzed the similarity and differences in amino acid sequences and expression patterns between birch BpSWEWET1c and several plants SWEET genes, such as CsSWEET1a, CsSWEET16, and AtSWEET16. Further validated the reliability of the inference (line461-463,471-474 in the revised manuscript).

Point 3: Is the RNA analyzed in Figure 6B root-derived or leaf-derived?

Response: Thank you for your question. The RNA-Seq data comes from leaf and has been explained in the paper (line137 in the revised manuscript).

Round 2

Reviewer 1 Report

The manuscript has been improved according to suggestion made.

Now It is worth publication.

The manuscript still needs  language correction.

Reviewer 3 Report

Manuscript An insight of Betula platyphylla SWEET gene family through Genome-wide identification, expression profiling and function analysis of BpSWEET1c under low temperature stress by Hao Zhang et al. has been significantly improved, problems have been fixed.

This work has been done carefully and has prospects for development.

Once corrected, it may be published as presented.

I recommend improving the resolution of the pictures.

Reviewer 4 Report

This manuscript has been revised according to the reviewer’s comments. I am satisfied with the revisions that have been made by the authors. I have nothing to comment on.